# Transcriptome Analysis Unravels Key Factors Involved in Response to Potassium Deficiency and Feedback Regulation of K^+^ Uptake in Cotton Roots

**DOI:** 10.3390/ijms22063133

**Published:** 2021-03-19

**Authors:** Doudou Yang, Fangjun Li, Fei Yi, A. Egrinya Eneji, Xiaoli Tian, Zhaohu Li

**Affiliations:** 1State Key Laboratory of Plant Physiology and Biochemistry, China Agricultural University, Beijing 100193, China; yddwx1991520@163.com (D.Y.); lizhaohu@cau.edu.cn (Z.L.); 2College of Agronomy and Biotechnology, China Agricultural University, Beijing 100193, China; lifangjun@cau.edu.cn (F.L.); yifei56@cau.edu.cn (F.Y.);; 3Department of Soil Science, Faculty of Agriculture, Forestry and Wildlife Resources Management, University of Calabar, Calabar 540271, Nigeria; aeeneji@yahoo.co.uk

**Keywords:** cotton, grafting, potassium deficiency, nutrient transporter, transcription factor

## Abstract

To properly understand cotton responses to potassium (K^+^) deficiency and how its shoot feedback regulates K^+^ uptake and root growth, we analyzed the changes in root transcriptome induced by low K^+^ (0.03 mM K^+^, lasting three days) in self-grafts of a K^+^ inefficient cotton variety (CCRI41/CCRI41, scion/rootstock) and its reciprocal grafts with a K^+^ efficient variety (SCRC22/CCRI41). Compared with CCRI41/CCRI41, the SCRC22 scion enhanced the K^+^ uptake and root growth of CCRI41 rootstock. A total of 1968 and 2539 differently expressed genes (DEGs) were identified in the roots of CCRI41/CCRI41 and SCRC22/CCRI41 in response to K^+^ deficiency, respectively. The overlapped and similarly (both up- or both down-) regulated DEGs in the two grafts were considered the basic response to K^+^ deficiency in cotton roots, whereas the DEGs only found in SCRC22/CCRI41 (1954) and those oppositely (one up- and the other down-) regulated in the two grafts might be the key factors involved in the feedback regulation of K^+^ uptake and root growth. The expression level of four putative K^+^ transporter genes (three *GhHAK5s* and one *GhKUP3*) increased in both grafts under low K^+^, which could enable plants to cope with K^+^ deficiency. In addition, two ethylene response factors (ERFs), *GhERF15* and *GhESE3*, both down-regulated in the roots of CCRI41/CCRI41 and SCRC22/CCRI41, may negatively regulate K^+^ uptake in cotton roots due to higher net K^+^ uptake rate in their virus-induced gene silencing (VIGS) plants. In terms of feedback regulation of K^+^ uptake and root growth, several up-regulated DEGs related to Ca^2+^ binding and CIPK (CBL-interacting protein kinases), one up-regulated *GhKUP3* and several up-regulated *GhNRT2.1s* probably play important roles. In conclusion, these results provide a deeper insight into the molecular mechanisms involved in basic response to low K^+^ stress in cotton roots and feedback regulation of K^+^ uptake, and present several low K^+^ tolerance-associated genes that need to be further identified and characterized.

## 1. Introduction

Potassium (K^+^) is an essential macronutrient that plays crucial roles in plant growth, development, and response to stress, which include balancing the electrical charge of membranes, activating enzymes, transporting of minerals and metabolites, turgor regulation, osmoregulation, as well as signal transduction [1,2]. Potassium starvation can inhibit plant photosynthesis, reduce transpiration rate and stomatal conductance, affect carbohydrate translocation and metabolism, weaken resistance to biological and abiotic stresses, and consequently decrease crop yield and quality [1,3]. Cotton (*Gossypium hirsutum* L.) is a potassium-sensitive crop due to its sparse roots and its relatively inefficient K^+^ absorption [3]. However, K^+^ availability in the soil is often limited which causes low-K^+^ stress during cotton growth periods [2,3].

The K^+^ uptake in plant roots involves diverse K^+^ channels and transporters. Plant K^+^ channels consist of members of the Shaker, tandem-pore (TPK), and two-pore channel (TPC) families, and K^+^ transporters include members of the KUP/HAK/KT, high-affinity K^+^ transporter (HKT), and cation–proton antiporter (CPA) families [4,5]. Under high external K^+^ concentrations (>0.3 mM), K^+^ channels mainly mediate low-affinity K^+^ uptake, whereas K^+^ transporters mainly mediate high-affinity K^+^ uptake under low external K^+^ concentrations (<0.2 mM) [4].

Plants have evolved complex signaling pathways to address the issue of fluctuating external K^+^ levels. Root cells could perceive external K^+^ deficiency and generate initial K^+^ signaling, and this K^+^ signaling is subsequently transduced or encoded by Ca^2+^ and reactive oxygen species (ROS) sensors, which then induce changes in signaling components, including phytohormones and transcription factors (TFs). Phytohormones and TFs then regulate the downstream (including K^+^ channels and transporters) transcriptional, translational, and posttranslational responses, and finally, plants exert many morphological and physiological adaptive changes that assist survival under K^+^-deficiency stress [5,6,7].

In higher plants, there are several types of “Ca^2+^ sensors” that decode Ca^2+^ at different levels in the cell [8]. For instance, calmodulin (CaM) and calmodulin-like (CML) can regulate functions of TFs. CaM binding proteins can bind to CaM in a Ca^2+^-dependent and reversible manner. Calcineurin B-like (CBL) protein is without enzymatic activity per se, but interacts with specific CBL-interacting protein kinases (CIPKs) to phosphorylate target proteins [9]. Calcium-dependent protein kinases (CDPKs or CPKs) have the unique feature of a catalytic protein kinase domain and a regulatory domain [10]. At the cellular level, K^+^ deficiency has been shown to elevate [Ca^2+^]_cyt_ that activates downstream signalling through CBL-CIPK complexes [11]. The CBL1/CBL9-CIPK23 complex can activate both K^+^ channel AKT1 [12] and K^+^ transporter HAK5 through phosphorylation to enhance K^+^ uptake [13]. Another K^+^ channel AKT2 mediates K^+^ transport in the phloem and regulates sucrose loading [14]. Its translocation from endoplasmic reticulum (ER) membrane to the plasma membrane (PM) is Ca^2+^-dependent and regulated by the CBL4 together with the CIPK6 in *Arabidopsis* [15].

Nicotinamide adenine dinucleotide phosphate (NADPH) oxidase, encoded by *Respiratory burst oxidase homologue* (*Rboh*) gene, is the main source of ROS production induced by stress (including low-K^+^ stress) [16]. Also, other oxidases and peroxidases are involved in ROS production in response to K^+^ deficiency [5]. To alleviate the damage caused by oxidative stress, plants have developed effective enzymatic defense system such as peroxidase (POD), glutathione S-transferase (GST), superoxide dismutase (SOD), peroxiredoxin (PRX), glutaredoxin (GRX), catalase (CAT), and ascorbate peroxidase (APX) [17], as well as non-enzymatic antioxidants, including glutathione (GSH), and ascorbate (AsA) [18].

There are evidences that K^+^ transporters from the KT/KUP/HAK family, may function as auxin carriers or are involved in auxin signaling. The alteration of root growth and development of K^+^-deprived plants was linked to decreased auxin concentrations and disruption of auxin distribution [19,20,21,22,23]. Potassium deprivation decreased cytokinin (CTK) level, which allows fast and effective stimulation of ROS accumulation, root hair growth and *HAK5* expression, leading to plant adaptation to low K^+^ conditions [24]. In contrast to CTK, the content of abscissic acid (ABA) increased under K^+^ deficiency [25], and the putative components of ABA signaling pathway may regulate the activity of K^+^ channels or transporters via phosphorylation /dephosphorylation. For example, the C terminal cytosolic region of K^+^ transporters KUP6 can be phosphorylated by the SNF1-related protein kinases 2E (SRK2E) [21]. In addition, the K^+^ channel AKT1 can be dephosphorylated by AKT1-interacting PP2C1 (AIP1) [26,27,28] or be inactivated by PP2CA through inhibiting the kinase activity of CIPK6 [12]. Ethylene signaling is a component of the plant’s response to low K^+^ that stimulates the production of ROS and induces the expression of high-affinity K^+^ transporter *HAK5* [29]. Ethylene signaling is also involved in the inhibition of lateral roots and improvement of root hair elongation in *Arabidopsis* under low K^+^ conditions [29,30]. Our previous work also showed that the cotton lateral root formation were significantly inhibited by low K^+^, which was accompanied by a strong increase (near six-fold) in ethylene release [19]. Furthermore, jasmonic acid (JA) participates in the response to K^+^ deficiency in *Arabidopsis* [31], rice [32], and wheat [25], and may have putative roles in nutrient storage and remobilization as well as enhancing the defense potential of K^+^-deficient plants [31,33].

Previous reports showed that the genes encoding K^+^ transporters, especially members of the KUP/HAK/KT family were mostly identified in response to K^+^ deficiency or deprivation [34,35,36,37]. The *HAK5* genes under K^+^ deprivation were strongly induced in roots of *Arabidopsis* (*AtHAK5*) [38], pepper (*CaHAK5*) [39], tomato (*LeHAK5*) [40], and cotton (*GhHAK5*) [41]. However, the details of how *HAK5* is regulated are still far from clear. Curently, only two TFs, AtRAP2.11 and AtARF2 that can bind to *AtHAK5* promoter and positively or negatively regulate *AtHAK5,* have been conclusively identified [42,43].

In the last two decades, a number of authors have focused on low-K^+^ stress using transcriptome, metabolome and proteomics analysis in model plants *Arabidopsis* and rice as well as other species [44,45,46,47]. However, there are no such reports on cotton yet to our knowledge. We once performed a reciprocal grafting using a K^+^ inefficient cotton variety, CCRI41 and a K^+^ efficient variety, SCRC22, and found that SCRC22 scion can feedback facilitate K^+^ uptake in CCRI41 rootstock under K^+^ deficiency [41]. Nevertheless, the underlying molecular mechanisms remain unclear. Here, we conducted a root transcriptome profiling of CCRI41/CCRI41 (scion/rootstock, self-grafts) and SCRC22/CCRI41 (reciprocal grafts) following three days of K^+^ deficiency (0.03 mM K^+^). The genes that were similarly regulated in self- and reciprocal grafts were considered as the basic response to low K^+^ conditions in cotton roots, and those only differently expressed in the roots of SCRC22/CCRI41 or oppositely regulated in CCRI41/CCRI41 and SCRC22/CCRI41 were regarded as specific response to low K^+^-stress in SCRC22/CCRI41, which would explain the feedback regulation of K^+^ uptake and root growth in cotton.

## 2. Results

### 2.1. Effects of Low-K^+^ Stress on Phenotype, Chlorophyll Content, Biomass, and K^+^ Content of CCRI41/CCRI41 and SCRC22/CCRI41

Grafting was performed hypocotyl-to-hypocotyl (Figure 1a) when the cotyledons of rootstock just fully expanded. The low-K^+^ stress (LK, 0.03 mM K^+^) was applied at three-leaf stage. After 16 days, we observed that the interveinal chlorosis on the third and fourth leaves was more severe in self-grafts of CCRI41/CCRI41 than in reciprocal-grafts of SCRC22/CCRI41 (Figure 1b). Consistently, CCRI41/CCRI41 showed lower chlorophyll content in these two leaves under low K^+^ condition (Figure 1c). Moreover, the biomass of roots, stem and leaves reduced to varying extents under K^+^ deficiency, and CCRI41/CCRI41 was impacted more strongly than SCRC22/CCRI41, especially in terms of root biomass (Figure 1d). The root-shoot ratio of CCRI41/CCRI41 decreased by 3.1% under low K^+^ condition, which was similar to previous reports [41]. However, the root-shoot ratio of SCRC22/CCRI41 increased by 3.3% (Figure 1e). Furthermore, the SCRC22/CCRI41 showed higher K^+^ concentration and accumulation in roots and leaves than CCRI41/CCRI41 under low K^+^ stress, and stored more K^+^ in stem than CCRI41/CCRI41 mainly due to higher dry weight (Figure 1f,g). These results suggest that SCRC22 scion could feedback enhance root growth and K^+^ uptake in CCRI41 rootstock.

### 2.2. Transcriptome Profiling of CCRI41/CCRI41 and SCRC22/CCRI41 in Response to K^+^ Deficiency

To investigate the changes of gene expression patterns involved in response to K^+^ deficiency and feedback regulation of K^+^ uptake and root growth in cotton, the roots were sampled after three days of low K^+^ treatment (LK, 0.03 mM K^+^) to perform comparative transcriptome analysis. At this time, the expression of the marker gene *GhHAK5* (*GhD_01G1760*) under K^+^ deficiency significantly differed between CCRI41/CCRI41 and SCRC22/CCRI41 (Appendix A).

An overview of the sequence assembly after Illumina sequencing is shown in Appendix A. A total of 719 million raw reads (697 million clean reads) were generated from 12 root samples. The Q30 (>93%) values indicated that the quality of sequencing data was sufficient to support further transcriptome analysis. An average ~96% of these high-quality reads were mapped to the cotton genome (*G. hirsutum* TM-1(AD)_1_) [48]. The relationship of transcriptome samples for CK (2.5 mM K^+^) and LK (0.03 mM K^+^) treatments was assessed by the global hierarchical clustering (Figure 2a). To further validate the quality of the gene activity profiles, eight genes were randomly selected to compare their Fragments Per Kilobase Million Mapped Reads (FPKM ) values and RT-qPCR data. The results showed that the difference in their relative expression levels evaluated by RT-qPCR between CK and LK were consistent with the difference in FPKM value (Appendix A), suggesting the data were appropriate for subsequent analysis.

The differentially expressed genes (DEGs) induced by low K^+^ condition were examined by comparing the expression levels between LK and CK using *p*-value < 0.05, and |log_2_ (Fold Change) | ≥1. Overall, 1968 and 2539 DEGs were identified in roots of CCRI41/CCRI41 (LR_41/41__CR_41/41_) and SCRC22/CCRI41 (LR_22/41__CR_22/41_), respectively (Figure 2b). A total of 585 DEGs were common in LR_41/41__CR_41/41_ and LR_22/41__CR_22/41_, named R common group (Figure 2c). In addition, 1954 DEGs were only identified in LR_22/41__CR_22/41_, named R_22/41_ unique group (Figure 2c). The DEGs in R common group except those oppositely induced (192) in LR_41/41__CR_41/41_ and LR_22/41__CR_22/41_ could be regarded as the basic response to K^+^ deficiency in cotton roots, whereas the DEGs in R_22/41_ unique group and those DEGs oppositely induced in LR_41/41__CR_41/41_ and LR_22/41__CR_22/41_ might be the key factors involved in the feedback regulation of K^+^ uptake and root growth.

### 2.3. Gene Function Enrichment Analysis of DEGs in Response to K^+^ Deficiency

Gene Ontology (GO) analysis was conducted with the adjusted FDR < 0.05 as significant enrichment. The DEGs in R common and R_22/41_ unique group were significantly enriched into 75 and 40 categories, respectively (Appendix A). For cellular components, the extracellular region (GO: 0005576) was enriched in both groups, and the ubiquitin ligase complex (GO: 0000151) was only enriched in R_22/41_ unique group. For the molecular functions category, the oxidoreductase activity (GO: 0016491), heme binding (GO: 0020037), electron carrier activity (GO: 0009055), catalytic activity (GO: 0003824) and nucleic acid binding transcription factor activity (GO: 0001071) were significantly enriched in both groups. For the biological processes category, the metabolic process (GO: 0008152) involved in the most abundant genes, and oxidation-reduction process (GO: 0055114) was the highest enrichment pathway in both groups (Appendix A).

The results of the Kyoto Encyclopedia of Genes and Genomes (KEGG) pathway analysis showed that biosynthesis of secondary metabolites, nitrogen metabolism, galactose metabolism and glucosinolate biosynthesis were significantly enriched (*q-*value < 0.05) in R common and R_22/41_ unique group (Appendix A), consistent with the fact that K^+^ affects the activity of many enzymes and several metabolic pathways [49]. In addition, plant hormone signal transduction and MAPK signaling pathway were only enriched in the R common group, while ABC transporters was only enriched in the R_22/41_ unique group (Appendix A).

### 2.4. DEGs Associated with Ca^2+^, ROS, and Phytohormone Signaling Pathway in Response to K^+^ Deficiency

For the R common group, 10 DEGs related to Ca^2+^ binding and eight DEGs associated with CaM binding were identified, but only two putative Ca^2+^ channel genes and one *GhCIPK* gene were found. In addition, five DEGs related to Ca^2+^ binding protein were up-regulated, and five DEGs related to CaM binding protein were down-regulated in the roots of both CCRI41/CCRI41 and SCRC22/CCRI41 (Figure 3a), suggesting that Ca^2+^ sensors may play important roles in response to K^+^ deficiency.

Three DEGs, including one annotated by Ca^2+^ channel activity (*Gh_D11G1884*), one related to Ca^2+^ binding (*Gh_A05G1742*) and one associated with CaM binding (*Gh_A07G2285*) were up-regulated in the roots of CCRI41/CCRI41 but down-regulated in the roots of SCRC22/CCRI41 (Figure 3a). Moreover, there were 55 Ca^2+^ related DEGs identified in SCRC22/CCRI41 alone. Among these, several related to Ca^2+^ and CaM binding, and *GhCIPKs* were up-regulated under low K^+^, which was notable compared with those numerous down-regulated DEGs related to Ca^2+^ signaling in R_22/41_ unique group (Figure 3b,c).

A total of eight DEGs related to ROS were identified in the R common group, including six putative *GhPOD* (*peroxidase*) genes, one putative *GhGST* (*glutathione S-transferase*) gene and one putative *GhPRX* (*peroxiredoxin*) gene (Figure 4a). Among the six DEGs associated with *GhPOD*, three of them were down-regulated in both grafts, and another three DEGs were oppositely regulated in two grafts (Figure 4a).

For the R_22/41_ unique group, one *GhRbohA* (*Gh_A02G1791*) and one *GhRbohD* (*Gh_D05G2471*) were significantly down-regulated (Figure 4b). Also, 24 out of 30 DEGs related to *GhPOD*, and eight putative *GhGST* DEGs genes were significantly down-regulated (Figure 4b,c).

The 19 and 16 DEGs related to phytohormone were found in R common and R_22/41_ unique group, respectively (Figure 5). The similarly regulated (both up- or both down-) DEGs in both grafts were associated with auxin (one up-regulated *GhSAUR* gene), CTK (one down-regulated *GhARR12* gene, and two up-regulated *GhARR18* genes), ABA (one down-regulated *GhSNRK2* gene, and one up-regulated *GhPP2C8* gene), ethylene (five down-regulated genes, *GhETR2*, *GhEBF1*, two *GhERF1B* genes, and *GhERF15A*), and BR (five down-regulated *GhTCH4* genes) (Figure 5a).

The one DEG each associated with auxin, CTK, ABA and SA were oppositely regulated (one up- and the other down-) in CCRI41/CCRI41 and SCRC22/CCRI41 (Figure 5a). For the R_22/41_ unique group, the DEGs related to auxin (*GhIAA9* and *GhGH3*), CTK (*GhARR2*, *GhARR12* and *GhARR9*), ethylene (*GhERF15D* and *GhERF1B*), BR (two *GhBKI1* genes), and JA (four *GhJAZ* genes) were significantly down-regulated, whereas the two auxin-responsive genes (*GhGH3.17* and *GhSAUR*) and one ABA receptor gene (*GhPYL*) were significantly up-regulated (Figure 5b).

### 2.5. DEGs Associated with Transporters in Response to K^+^ Deficiency

The K^+^ uptake, transportation, and distribution are primarily mediated by K^+^ transporters and K^+^ channels located in the plasma membrane [49]. Therefore, we focused on the genes related to ion transmembrane transporter activity (Figure 6a, Appendix A). Unexpectedly, only three *GhHAK5* (*Gh_A01G1516*, *Gh_D01G1760* and *Gh_D01G1763*) and one *GhKUP3* (*Gh_D04G0700*), all from the KUP/HAK/KT family, were significantly up-regulated in the roots of both CCRI41/CCRI41 and SCRC22/CCRI41 after three days of low-K^+^ stress (Figure 6a, Appendix A).

In addition, only one *GhKUP3* gene (*Gh_A05G2905*) was significantly up-regulated in R_22/41_ unique group, which was even coupled with the down-regulation of a pair of *GhHAK5* genes (*Gh_A12G0074* and *Gh_D12G0090*) (Figure 6b). Considering the higher K^+^ uptake activity of SCRC22/CCRI41 relative to CCRI41/CCRI41 under low K^+^ condition, such a few specifically up-regulated DEGs in SCRC22/CCRI41 was unexpected.

Among the total of five *GhHAK5s* regulated by low K^+^, only the *GhHAK5* (*Gh_D01G1760*), characterized in our previous study [41], showed higher FPKM value (>16), whereas other *GhHAK5*s had lower FPKM values (<4.6) (Appendix A). Also, the FPKM value of two up-regulated *GhKUP3* genes (*Gh_D04G0700* and *Gh_A05G2905*) was even higher than that of *GhHAK5* (*Gh_D01G1760*) (Appendix A), indicating their important roles in cotton.

Besides K^+^ transporter genes, the expression level of other nutrients transporter genes altered in cotton roots exposed to K^+^ deficient surroundings. For example, one putative Na^+^ transporter gene, one NO_3_^−^ transporter gene, one iron transporter gene, three zinc transporter genes, two copper transporter genes, and one boron transporter gene were all significantly down-regulated in the roots of CCRI41/CCRI41 and SCRC22/CCRI41 (Figure 6a). The overall nutrient demand is less or root to shoot translocation is reduced under low-K^+^ stress, which probably confers feedback inhibition of expression of related genes in roots [11].

In terms of specific DEGs in the roots of SCRC22/CCRI41, several putative NO_3_^−^ transporter genes were found to be predominantly up-regulated, especially two *GhNRT2.1* genes (*Gh_A03G0257* and *Gh_D03G1307*) (Figure 6b). In addition, another *GhNRT2.1* (*Gh_D08G2124*) was down-regulated in CCRI41/CCRI41 but up-regulated in SCRC22/CCRI41 under low-K^+^ stress. Also, one of the phosphate (Pi) transporter genes, *GhPHT1;5* (*Gh_A02G0203*), was specifically down-regulated in the roots of SCRC22/CCRI41, and another Pi transporter gene *GhPHT1;5* (*Gh_D10G1372*) was down-regulated in SCRC22/CCRI41 but up-regulated in CCRI41/CCRI41. Several aluminum (Al) and boron (B) transporter genes were only identified as up-regulated genes in SCRC22/CCRI41. The other three Al transporter genes (*Gh_D07G1355*, *Gh_A07G1264* and *Gh_D08G1184*) were up-regulated in SCRC22/CCRI41 but down-regulated in CCRI41/CCRI41. It appears that the down-regulation of Pi transporter genes and up-regulation of Al and B transporter genes may be involved in feedback regulation of K^+^ uptake in cotton roots.

Several *GhABC* genes were slight down-regulated, while two *GhABC6* (*Gh_A05G3075* and *Gh_D04G0567*) were strongly up-regulated (over five-fold) in the roots of SCRC22/CCRI41 under low K^+^ condition (Figure 6c). Furthermore, two genes related to aquaporin and one gene related to sugar transporter gene *GhSWEET17* (*Gh_A13G1540*) were up-regulated, while four aquaporin genes and one putative sugar transporter gene *GhSWEET2* (*Gh_D11G2975*) were down-regulated beyond the 1-fold cut-off in the roots of SCRC22/CCRI41 alone (Figure 6c).

### 2.6. DEGs Associated with Transcription Factors (TFs) in Response to K^+^ Deficiency

As indicated by GO analysis, the nucleic acid binding transcription factor activity (GO: 0001071) was enriched in roots under K^+^ deficiency. A total of 67 and 201 DEGs related to TFs were identified in R common and R_22/41_ unique group, respectively (Appendix A). These TFs belong to 30 families, and covered almost all types of TFs. For the R common group, the *ERFs* family had the most DEGs (17), followed by *NACs* (14), *MYBs* (8), and *WRKYs* (6). As shown in Appendix A, 51 TFs in the R common group were similarly regulated (five up-regulated and 46 down-regulated) by K^+^ deficiency in the roots of both grafts, and other 16 TFs were differently regulated in the roots of SCRC22/CCRI41 and CCRI41/CCRI41.

For the R_22/41_ unique group, the most abundant differential expression TFs also belonged to *ERFs* (41), followed by *MYBs* (30), *WRKYs* (25), *NACs* (18), *C2H2s* (17), and *bHLHs* (14) (Appendix A). Interestingly, 78.1% of the total 201 TFs in this group were down-regulated by low-K^+^ stress.

As *GhHAK5* (*Gh_D01G1760*) is a marker gene induced by K^+^ deficiency [41], the TF-binding sites were predicted from the putative promoter sequences (2 kb upstream from the transcriptional start site) of *GhHAK5* gene aiming to identify its candidate TFs from differently expressed TFs. For the R common group, four down-regulated TFs (*GhWRKY33*, *GhDOF1.5*, and a pair of *GhESE3*) were predicted to modulate the expression of *GhHAK5* (Figure 7a). For the R_22/41_ unique group, two *WRKYs* (*GhWRKY29* and *GhWRKY70*), three *NACs* (*GhNAC78*, *GhNAC86* and *GhNAC100*), and four *ERFs* (*GhERF5*, *GhERF13*, *GhERF105*, and *GhERF1B*) were predicted to be involved in the regulation of *GhHAK5* (Figure 7b). These nine TFs were down-regulated under K^+^ deficiency (Figure 7b).

### 2.7. GhERF15A and GhESE3A Are Probably Involved in Negatively Regulating K^+^ Uptake under K^+^ Deficiency

To further understand the roles of TFs in response to low K^+^, two *TFs* were selected to characterize the function. One of them was *GhERF15A* (*Gh_A08G1686*) that was involved in the ethylene signaling pathway (Figure 5a) and down-regulated in the roots of both CCRI41/CCRI41 (1.208-fold) and SCRC22/CCRI41 (2.423-fold) under low K^+^. The other *TF* selected was also a putative ethylene response factor *GhESE3A* (*Gh_A07G0251*) that was predicted as the TF of *GhHAK5* (Figure 7a).

The expression pattern of *GhERF15A* and *GhESE3A* as well as *GhHAK5* (*Gh_D01G1760*) was examined in the roots of SCRC22 (non-grafts) at the three-leaf stage. Compared with the plants with sufficient K^+^ (CK, 2.5 mM K^+^), the expression level of *GhERF15A* and *GhESE3A* strongly fluctuated over time under low K^+^ (0.03 mM K^+^), and showed a largely reversed pattern relative to *GhHAK5* (*Gh_D01G1760*) (Appendix A), suggesting their negative regulation of K^+^ uptake. In addition, the relative expression of *GhERF15A* and *GhESE3A* in the roots of *GhHAK5* (*Gh_D01G1760*) RNAi lines was significantly lower than that of wild type S3 (Appendix A).

We investigated the function of *GhERF15A* and *GhESE3A* using virus-induced gene silencing (VIGS) assay in the variety of SCRC22. As shown in Figure 8a,b, a pair of *GhERF15* (*Gh_A08G1686* and *Gh_D08G2044*) and a pair of *GhESE3* (*Gh_A07G0251* and *Gh_D07G0308*) were effectively silenced, and their expression levels decreased by 62.4% and 42.7%, respectively. Seedlings with equal fresh weight were selected to determine the K^+^ uptake activity. As expected, VIGS-*GhERF15* and VIGS-*GhESE3* seedlings showed 38.0%–52.4% and 28.5%–105.8% higher net K^+^ uptake rate than VIGS-Ctrl plants (Figure 8c,d).

## 3. Discussion

Compared with nitrogen (N), phosphorus (Pi), iron (Fe), and sulphate (S), the number of DEGs under low-K^+^ stress was less in *Arabidopsis* and other plants [31,34,50]. However, there is no information on transcriptome analysis in relation to N and other nutrients in cotton. Some reports showed that the quantity of DEGs under salt stress varied from 3800 to 22,000 in cotton [51,52]. Han et al. identified 4627 DEGs in cotton under cadmium (Cd) toxicity [53]. In the present study, around 2000 and 2500 DEGs (*p*-value < 0.05, and |log_2_ (Fold Change) | ≥ 1.) were identified in the roots of CCRI41/CCRI41 and SCRC22/CCRI41 following three days of K^+^ deficiency (Figure 2b). The smaller number of DEGs in response to K^+^ deficiency indicates that post-transcriptional regulation and epigenetic modification are also important for plants to respond to low K^+^ stress. Moreover, the number of up- and down-regulated DEGs in CCRI41/CCRI41 was similar, but the number of down-regulated DEGs in SCRC22/CCRI41 was nearly two-fold those up-regulated (Figure 2b), suggesting that the down-regulated of some negative regulators might be important for SCRC22 scion to enhance K^+^ uptake and root growth of CCRI41 rootstock.

### 3.1. Ca^2+^ and ROS Signaling Involved in the Response to K^+^ Deficiency and Feedback Regulation of K^+^ Uptake and Root Growth

The elevation of [Ca^2+^]_cyt_ was elicited by low-K^+^ stress in a distinct spatial and temporal pattern, which could be subsequently decoded by various Ca^2+^ sensor, and then trigged the downstream responses [54]. In this study, the expression level of several putative Ca^2+^ channel genes specifically reduced in the roots of reciprocal grafts of SCRC22/CCRI41 under low K^+^ condition (Figure 3b), indicating the Ca^2+^ signal in CCRI41 rootstock was likely attenuated by SCRC22 scion. In contrast to the Ca^2+^ channel genes, several genes related to Ca^2+^ binding protein and *GhCIPK* genes, including *GhCIPK23* (*Gh_D09G1024*), were only induced in the roots of SCRC22/CCRI41 under K^+^ deficiency (Figure 3b,c), suggesting the sensitivity of Ca^2+^ signaling system may be feedback enhanced by SCRC22 scion. The CIPK23 has been reported to phosphorylate and activate K^+^ channel AKT1 and K^+^ transporter HAK5 in *Arabidopsis* [13,27]. Therefore, the increased expression level of *GhCIPK23* may participate in the feedback regulation of K^+^ uptake by modulating the activity of the K^+^ transporters, not excluding K^+^ channels under K^+^ deficiency.

By adding either H_2_O_2_ (a major type of ROS) or DPI (NADPH oxidase blocker) and using the *rhd2* mutant, it was found that H_2_O_2_ plays a role in controlling the expression of certain genes in response to K^+^ deprivation [55]. For instance, ROS can directly regulate *AtHAK5* expression either through the TF RAP2.11 (related to AP2 11) [42] or in an ethylene-dependent manner [56]. Recently, Huang et al. demonstrated that RBOHD-mediated transcriptional and post-translational activation of plasma membrane H^+^-ATPase operated upstream of K^+^ uptake transporters HAK5 [57]. However, the specificity of the ROS-induced responses are still not well known, in part because little is known about nutrient signaling cascades in plants [56]. In our study, only a few of DEGs related to ROS, including three down-regulated *POD* genes, one down-regulated *PRX* gene, and one up-regulated *GST* gene, were identified in the roots of both grafts under K^+^ deficiency (Figure 4a), suggesting that ROS signaling and metabolism may be inactive at the time point we studied [three days after low-K^+^ (0.03 mM) stress]. In addition, the expression of *GhRbohA* (*Gh_A02G1791*) and *GhRbohD* (*Gh_D05G2471*) were inhibited in SCRC22/CCRI41 alone under K^+^ deficiency (Figure 4b). In general, ROS not only activates Ca^2+^ permeable channels but also results in K^+^ efflux from root cells [58]. We consider that the down-regulated *GhRbohA* and *GhRbohD* may be beneficial for K^+^ retention in the roots of SCRC22/CCRI41 exposed to K^+^ deficiency by inhibiting K^+^ loss, and thus leading to a higher root K^+^ content compared with CCRI41/CCRI41. Consistent with the down-regulation of *GhRbohA* and *GhRbohD*, most of the scavenging ROS genes were down-regulated in SCRC22/CCRI41 under low-K^+^ stress. It perhaps make sense for root growth since the inactive ROS production and elimination may save energy and assimilate.

### 3.2. Phytohormone Signaling Involved in Response to K^+^ Deficiency and Feedback Regulation of K^+^ Uptake and Root Growth

Previous transcriptome studies in rice, tomato, and soybean showed that dozens of DEGs related to auxin responded to K^+^ deficiency [34,59,60]. However, in the present study, only a *GhSAUR* gene (*Gh_A08G1223*) was induced by low K^+^ in the roots of both grafts. Although the *SAUR* (*small auxin-up RNA)* is the largest family of early auxin responsive genes in higher plants, the functions of only few *SAUR* genes are known owing to functional redundancy among the many family members [61]. The five DEGs related to auxin were specifically up- or down-regulated in SCRC22/CCRI41, including an up-regulated *GhGH3* gene (*Gh_D03G0465*). GH3 proteins are involved in various developmental processes and environmental responses in plants, by modulating the activities or availabilities of plant hormones [62]. In cotton, *GhGH3.5* may be involved in drought and salt tolerance [63]. Moreover, one *auxin efflux carrier* gene *GhPILS* (*Gh_A05G1716*) was up-regulated in the roots of CCRI41/CCRI41 but down-regulated in the roots of SCRC22/CCRI41. Considering the greatly different root biomass between the two grafts subjected to prolonged low-K^+^ stress as well as the important roles of auxin in regulating root growth, we hypothesize that the auxin-related DEGs specifically identified in SCRC22/CCRI41 and the *GhPILS* (*Gh_A05G1716*) partly participated in the feedback regulation of root growth.

Some putative components of ABA signaling have been involved in the response to K^+^ tolerance. PP2C inactivates K^+^ channel, and SRK2E kinase activates K^+^ transporter [12,21]. An ortholog of *AtPP2CA* was up-regulated in the roots of *M. truncatula* plants inoculated with Arbuscular mycorrhizal (AM) under low-K^+^ regime [64]. Proteomic data indicated that the abundance of two PP2Cs was significantly changed in K^+^-deficient wheat seedlings [25]. In this study, one *GhPP2C8* gene (*Gh_D03G0739*) was up-regulated, and one *GhSnRK2* gene (*Gh_D03G0754*) was down-regulated in the roots of both grafts, suggesting that ABA signaling may play a negative role in basic response to low K^+^ stress. However, it seems that SCRC22 scion enhanced the ABA signaling in CCRI41 rootstock under K^+^ deficiency, as indicated by the up-regulated ABA receptor gene *GhPYL* (*Gh_A12G1895*), and down-regulated ABA negative regulator *GhPP2C51*. We speculate that the activated ABA signaling may be associated with the feedback effects of scion on rootstock.

The ethylene pathway is deeply involved in K^+^ uptake and change of root system architecture under K^+^ deficiency [56]. The transcriptome results of Fan et al. indicated that the genes related to ethylene biosynthesis and signaling were involved in the response to low-K^+^ stress in watermelon roots [36]. Among these, several genes were down-regulated at 6 h of K^+^ deprivation [36], which differed with the results in *Arabidopsis* [56]. Moreover, no genes related to ethylene were found to respond to K^+^ deficiency in barley roots [35]. In the present study, we identified several down-regulated genes related to ethylene in the roots of both CCRI41/CCRI41 and SCRC22/CCRI41 under low K^+^ condition, including one receptor gene *GhERT*, one *GhEBF1/2* (a negative factor in ethylene pathway) gene, and three *GhERF1/2* genes. In terms of specifically regulated genes in SCRC22/CCRI41 roots, two *GhERFs* were down-regulated under low-K^+^ stress, and one of them, *GhERF15A* (*Gh_A08G1686*) is possibly associated with the inhibition of K^+^ uptake (see below). Therefore, its down-regulation may contribute to the feedback enhancement of K^+^ uptake under K^+^ deficiency. The differences in ethylene-related gene expression across plant species could be due to different signaling pathways, or more likely differences in physiological processes involved in response time to low K^+^ [56].

Jasmonic acid (JA) is involved in response to K^+^ deficiency in *Arabidopsis* [31], rice [32], and wheat [25]. In our study, four *GhJAZ* (*Jasmonate zim-domain,* repress the activity of various TFs) genes (*Gh_D02G1776*, *Gh_A03G1341*, *Gh_D05G0352,* and *Gh_A05G0260*) were specifically down-regulated in the roots of SCRC22/CCRI41, implying that JA pathway may be involved in the feedback regulation of K^+^ uptake in cotton. In addition, we found that BR signaling may be involved in the responses to K^+^ conditions in cotton roots, which is a novel finding that has not been reported yet. The five *GhTCH4* genes were down-regulated in the roots of both grafts subjected to low-K^+^ stress. The *GhTCH4* gene encodes a BR-regulated xyloglucan endotransgiycosylase (XETs) that plays a role in cell wall modifications in response to environmental stress and during morphogenesis [65]. Therefore, the *GhTCH4* genes are likely involved in response to K^+^ deficiency by affecting root cell wall properties in cotton. Furthermore, the expression of two *GhBKI1* (a negative regulator in BR signaling pathway) genes (*Gh_D01G1023* and *Gh_A01G0972*) were inhibited in SCRC22/CCRI41 roots alone by low K^+^, suggesting that SCRC22 scion elicited BR signaling in CCRI41 rootstock under low-K^+^ condition, which likely facilitate the downstream components of BR pathway and thus maintain the root growth of reciprocal grafts.

### 3.3. Transporters Involved in the Response to K^+^ Deficiency and Feedback Regulation of K^+^ Uptake

It was shown that although changes in K^+^ supply (either short- or long-term) affected the expression of a large number of membrane transporter genes, but surprisingly only a few genes belonged to K^+^ transporters that have been reported. Similar results were reported for *Arabidopsis* [31], wheat and rice [66], and model legume, *M. truncatula* [63]. Also, only three *GhHAK5* genes and one *GhKUP3* gene were induced by low K^+^ in the roots of both grafts in this study (Figure 6a). The transcripts of members of the KUP/HAK/KT family usually increased under K^+^ deficiency [34,36,37]. Moreover, genotypic differences in K^+^ efficiency were associated with the number and amplitude of up-regulated K^+^ transporter genes in cotton [41] and wheat [67]. Therefore, the up-regulated HAK-type transporters may be a rapid, direct and common strategy for plants to increase K^+^ uptake and overcome K^+^ deficiency.

Besides absorbing K^+^, the K^+^ transporters, from the KT/KUP/HAK family, also function as auxin carriers or are involved in auxin signaling [22,23]. Recently, it was found that OsHAK5 alters the architecture of rice root and shoot by regulating ATP-dependent transmembrane auxin transport [68], and the K^+^ transporter KUP9 maintains *Arabidopsis* root meristem activity and root growth by regulating K^+^ and auxin homeostasis in response to low-K^+^ stress [69]. Thus, the differently expressed *GhHAK5* genes and *GhKUP3* gene in this study may also regulate cotton root development by altering auxin distribution.

Grafting can change the mode of K^+^ absorption in tobacco root, from being dominated by high-affinity transport system (HATS) to being jointly responsible by HATS and low-affinity transport system (LATS) [70]. In this study, SCRC22 scion also caused some specific expression changes in K^+^ transporter genes in CCRI41 rootstock, including one up-regulated (2.14-fold) *GhKUP3* gene (*Gh_A05G2905*) and a pair of down-regulated *GhHAK5* genes (Figure 6b). Since the roots of SCRC22/CCRI41 had a significantly higher K^+^ uptake activity than CCRI41/CCRI41 under low K^+^ condition, we speculate that this GhKUP3 may act as an important K^+^ transporter in cotton roots, and participate in the feedback regulation of root K^+^ uptake. In *Arabidopsis*, the KUP7 is also a major component involved in K^+^ uptake besides AKT1 and HAK5 [71].

The expression of *Nitrate transporter 1/peptide transporter* (*NRT1/PTR*) genes regulated by external potassium has been reported for *Arabidopsis* [31], rice [34], wheat [67], banana [47], and pear [37]. Consistently, we identified several up-regulated *GhNRT2.1s* in the roots of SCRC22/CCRI41 (Figure 6a, b). This is probably because K^+^ and nitrate (NO_3_^−^) intersect at multiple levels, including uptake, root to shoot transport, and cellular metabolism [72]. Moreover, the recent study by Li et al. revealed that the NRT1.5/NPF7.3 can also function as a proton-coupled H^+^/K^+^ antiporter, which mediates root-to-shoot K^+^ translocation [73]. Another nitrate transporter1 member, NRT1.1/NPF6.3, acts as a coordinator in K^+^ uptake and root-to-shoot allocation [74]. Therefore, it seems that the nitrate transporter genes identified in our results could be directly or indirectly involved in the feedback regulation of K^+^ uptake in cotton roots.

The Aluminum activated malate transporter 1 (ALMT1) was found to inhibit root cell elongation by mediating malate exudation under Pi deprivation [75]. In this study, *GhALMT10*, *GhALMT14,* and *GhALMT8* were up-regulated only in roots of SCRC22/CCRI41 under K^+^ deficiency (Figure 7a,b), which remind that ALMT might be also involved in the feedback regulation of root growth in SCRC22/CCRI41.

There is limited literature on plant sugar transport in response to K^+^ deficiency [58]. In our study, two putative sugar transporter genes *GhSWEET17* (*Gh_A13G1540*) and *GhSWEET2* (*Gh_D11G2975*) differently expressed in the roots of SCRC22/CCRI41 alone under low K^+^ condition; *GhSWEET17* was up-regulated and *GhSWEET2* was down-regulated (Figure 6c). In *Arabidopsis*, AtSWEET17 mediates fructose transport across the tonoplast of roots and leaves [76]. AtSWEET2 is localized in the tonoplasts of root cells, particularly in the cortex and epidermis, to play a role in vacuolar glucose sequestration, thereby limiting the efflux of carbon from roots [77]. Considering the obviously bigger roots of SCRC22/CCRI41 relative to CCRI41/CCRI41 under low-K^+^ stress, sugar transport may be involved in feedback enhancement of root growth by regulating assimilate distribution among organs.

### 3.4. Transcription Factors Involved in the Response to K^+^ Deficiency and Feedback Regulation of K^+^ Uptake and Root Growth

Transcription factors (TFs) may act as molecular switches to regulate clusters of gene expression in response to stress in higher plants [78]. It has been known that the Phosphate starvation response 1 (PHR1) is the key TF in Pi starvation; it coordinates with other MYBs and WRKYs to up- or down-regulate a subset of *Pi starvation-induced* (*PSI)* genes in response to Pi deficiency [79]. In addition, there is a key TF, NIN-like 7 protein (NLP7), and a series of other TFs implicated in the control of genes related to NO_3_^−^ transport and metabolism under low NO_3_^−^ conditions [80]. However, only a few TFs have been found to directly regulate the expression of K^+^ transporter genes so far. Besides AtRAP2.11 and AtARF2 that were known positively and negatively regulating *AtHAK5* respectively in *Arabidopsis*, four other TFs, *Dwarf and Delayed Flowering* 2 (*DDF2*), *Jagged Lateral Organs* (*JLO*), *Transcription initiation Factor II_A gamma chain* (*TFII_A*), and *basic Helix–Loop–Helix 121* (*bHLH121*), binding to the *HAK5* promoter to activate *HAK5* expression, were isolated via activation tagging of the *AtHAK5* promoter fused with a luciferase reporter gene [81]. Recently, the TF MYB59 was found to bind directly to the promoter of *NPF7.3* (a proton-coupled H^+^/K^+^ antiporter, mediates root-to-shoot K^+^ translocation) and respond to low-K^+^ stress by regulating the expression of *NPF7.3* in *Arabidopsis* roots [71]. In the case of rice, OsMYBc has been identified using a yeast one-hybrid approach, which can bind to the *OsHKT1;1* promoter and then activate its transcription [82]. In addition, the overexpression of *SiMYB3* from foxtail millet can promote the root elongation and improve K^+^ deficiency tolerance in transgenic *Arabidopsis* [83].

A wide array of transcriptome studies found dozens to hundreds of differently expressed TFs in response to K^+^ deficiency in plants, and MYB, ERF and WRKY families were most commonly identified [37,47,59,65,84], suggesting their implications in coping with external K^+^ deprivation. In this study, we identified 67 and 201 DEGs encoding TFs in the roots of both grafts and only in the roots of SSCRC22/CCRI41 under K^+^ deficiency (Appendix A). The members of ERF family were the most numerous either in both grafts or in SCRC22/CCRI41 alone, which was followed by NAC and MYB families in the former or MYB and WRKY families in the latter. Therefore, it could be supposed that the transcription regulation strategy of feedback improvement of K^+^ uptake was different from that of basic/common response to low K^+^ in cotton roots.

Only 10–20% differently expressed TFs were up-regulated by low K^+^ in our study, and majority was down-regulated, suggesting the TFs respond to low-K^+^ stress probably through relief of certain transcription inhibitions in cotton roots. Some previous reports showed similar results [36].

Importantly, we preliminarily characterized the function of two ERF TFs, GhERF15 and GhESE3, in the present study, and found that they negatively regulate K^+^ uptake in cotton roots. Although GhESE3A was predicted to bind to the promotor of *GhHAK5* (*Gh_D01G1760*), the results of dual luciferase reporter assay in cotton protoplast indicated that both GhERF15A and GhESE3A cannot significantly regulate the expression of *GhHAK5*. Since the nature of plant transcription regulatory network is complex and redundant [85], we guess that GhERF15A and GhESE3A may regulate *GhHAK5* by forming complexes with other proteins, or directly regulate other target genes to participate in the K^+^ absorption in cotton roots.

Future work will further characterize the functions of the two selected TFs above and other important candidate genes, and will screen more candidate genes in response to K^+^ limitation and involved in feedback regulation of K^+^ uptake and root growth by using SCRC22 self-grafts (SCRC22/SCRC22) and another reciprocal-graft CCRI41/SCRC22 besides CCRI41/CCRI41 and SCRC22/CCRI41. Also, the dynamic transcriptome patterns over time will be explored to better understand the molecular mechanisms underlying K^+^ uptake and its regulation pathways.

## 4. Materials and methods

### 4.1. Plant Materials and Growth Condition

A K^+^ efficient cotton variety, SCRC22 (developed by the Cotton Research Center, Shandong Academy of Agricultural Sciences) and a K^+^ inefficient variety, CCRI41 (developed by the Cotton Research Institute, Chinese Academy of Agricultural Sciences) were selected as materials. The self-grafts of CCRI41 (CCRI41/CCRI41, scion/rootstock) and reciprocal grafts SCRC22/CCRI41 were used for phenotype determination and RNA-Seq (RNA sequencing) analysis. When the cotyledons of rootstock just fully expanded (the scion being four days younger), the grafting was performed hypocotyl-to-hypocotyl as described previously [86].

In addition, SCRC22 (non-grafts) and the RNAi lines of K^+^ transporter *GhHAK5* (*Gh_D01G1760*) were used to show the expression pattern of some candidate genes, and the function of candidate genes was characterized in SCRC22 via VIGS (virus-induced gene silencing) assay.

The hydroponic experiments were conducted in a growth chamber under the following conditions: 14 h light/10 h dark photoperiod, 28 ± 2 °C day/22 ± 2 °C night temperatures, relative humidity 65% and 400 μmol·m^−2^·s ^−1^ photosynthetically active radiation. The modified Hoagland’s solution was used in this study, containing 2.5 mM KNO_3_, 2.5 mM Ca(NO_3_)_2_, 1 mM MgSO_4_, 0.5 mM (NH_4_)H_2_PO_4_, 0.1 mM Fe Na EDTA, and micronutrients (2 × 10^−4^ mM CuSO_4_, 1 × 10^−3^ mM ZnSO_4_, 2 × 10^−2^ mM H_3_BO_3_, 5 × 10^−6^ mM (NH_4_)_6_Mo_7_O_24_ and 1 × 10^−3^ mM MnSO_4_). During growth period of seedlings, the solutions were changed every four days, and deionized water was added daily to replace the water lost by evapo-transpiration. Oxygen was constantly provided to the solutions by air pump.

### 4.2. Photography of K^+^ Deficiency Symptoms and Measurement of Biomass, Chlorophyll and K^+^ Content

The grafts of CCRI41/CCRI41 and SCRC22/CCRI41 were raised in solutions containing 2.5 mM K^+^ until the three-leaf stage. Then, half of them was subjected to low K^+^ (LK, 0.03 mM K^+^), and the other half was continually cultured in normal condition (CK, 2.5 mM K^+^). After 16 days of K^+^ deficiency, seedlings were harvested. All leaves were photographed to present the effects of K^+^ limitation on leaf color and chlorosis. In addition, 0.2 g leaf disks were collected from the third, fourth and fifth true leaves to determine chlorophyll content following Tang et al [87]. Then, seedlings were separated into roots, stem and leaves, and dried at 80 °C. After weighing the dry weight, the K^+^ concentration of three subsamples from each organ were measured by an atomic absorption spectrophotometer (Z-2000, Hitachi, Japan) [88]. The K^+^ concentration times dry weight was K^+^ accumulation. Each treatment had three biological replicates, and each replicate included three plants.

### 4.3. RNA Isolation, Library Construction, and Sequencing

The grafted seedlings were subjected to low K^+^ (LK, 0.03 mM K^+^) at the three-leaf stage, with 2.5 mM K^+^ as control (CK). After three days of LK stress, the fine roots (diameter less than 1 mm) were sampled and immediately frozen in liquid nitrogen, then stored at −80 °C. Each treatment had three biological replicates, and each replicate was collected from five plants.

Total RNA was extracted and purified using the EASYspin Plus Complex Plant RNA Kit (Aidlab Biotech, Beijing, China). RNA purity was checked using the kaiaoK5500^®^ spectrophotometer (Kaiao, Beijing, China). RNA quality was assessed by Nano 6000 Assay Kit of the Agilent Bioanalyzer 2100 system (Agilent Technologies, Santa Clara, CA, USA). Thereafter, 3 µg high-quality total RNA was used as input material for library construction and Illumina sequencing.

Following the manufacturer’s recommendations, sequencing libraries were generated using NEBNext^®^ Ultra™ RNA Library Prep Kit for Illumina^®^ (#E7530L, NEB, Ipswich, MA, USA) and index codes were added to attribute sequences to each sample. The RNA concentration of library was measured using Qubit® RNA Assay Kit in Qubit® 3.0 to preliminarily quantify and then dilute to 1 ng/μL. Insert size was assessed using the Agilent Bioanalyzer 2100 system (Agilent Technologies, Santa Clara, CA, USA). After cluster generation, the library preparations were sequenced on the Illumina platform, with an average reading length of PE150. The library construction, and sequencing were performed by Anoroad Technologies (Beijing, China).

### 4.4. Read Mapping and Analysis of RNA-Seq

The clean reads were obtained by removing the junction sequences and low-quality sequences. The remaining reads were mapped to the *G. hirsutum* genome [48] using HISAT2 [89]. Reads count for each gene in each sample was done with HTSeq v0.6.0, and FPKM (Fragments Per Kilobase Millon Mapped Reads) was then calculated to estimate the expression level of genes in each sample [90]. DESeq2 (https://bioconductor.org/packages/release/bioc/html/DESeq2.html) (accessed on 8 December 2020) was used to estimate the expression level of each gene by the linear regression, then the *p*-value was calculated with Wald test. Finally, the *p*-value was corrected by the Benjamini and Hochberg method [91]. Genes with *p*-value < 0.05 and |log_2_ (Fold Change) | ≥ 1 were identified as DEG [91].

Sample relationships were analyzed by hierarchical clustering that was performed by the OmicShare (https://www.omicshare.com/tools/home/index/index.html) (accessed on 9 December 2020) with the HCA method.

### 4.5. Functional Annotation of DEGs

To further assign and annotate the DEGs, Gene ontology (GO) enrichment analysis (http://www.geneontology.org/) (accessed on 13 November 2020) and Kyoto Encyclopedia of Genes and Genomes (KEGG) pathway analysis (http://www.kegg.jp/) (accessed on 8 December 2020) were used to identify the genes in each library [92,93].

### 4.6. Reverse Transcription Quantitative Polymerase Chain Reaction (RT-qPCR) Analysis

To test the quality of RNA-seq sequencing, the full-length cDNA of randomly selected genes was synthesized using Oligo d (T) primer and M-MLV reverse transcriptase (Takara, Kusatsu, Japan) with 2 μg RNA. Then RT-qPCR was conducted in an Applied Biosystems 7500 Fast Real-Time PCR System (Applied Biosystems, Waltham, CA, USA) using SYBR^®^ Premix Ex Taq ™ (Takara, Kusatsu, Japan) following conditions: 95 °C for 30 s, 40 cycles of 95 °C for 5 s, 60 °C for 34 s, and 95 °C for 15 s, 60 °C for 60 s, then 95 °C for 30 s, finally 60 °C for 10 s. A melting curve was performed from 60 to 95 °C to check the specificity of the amplified product. *GhActin9* was used to normalize gene expression levels. The relative gene expression was calculated using the 2^−ΔΔCt^ method [94]. The primers for RT-qPCR are listed in Appendix A.

### 4.7. Prediction of TFs Binding to the Promoter of K^+^ Transporter GhHAK5 (Gh_D01G1760)

Putative promoter sequences (2 kb upstream from the transcriptional start site) of *GhHAK5* were searched in Cottongen (https://www.cottongen.org/retrieve/sequences) (accessed on 18 February 2021). Then, the prediction of TF-binding sites in this region was performed with the online software PlantRegMap (http://www.plantregmap.cbi.pku.edu.cn) (accessed on 18 February 2021) [95].

### 4.8. Detect the Transcription Pattern of Candidate Genes

Seedlings of SCRC22 (non-grafts) were exposed to low K^+^ (0.03 mM K^+^) at the three-leaf stage, with 2.5 mM K^+^ as control. Then the fine roots with diameter less than 1 mm were collected at 0, 6, 12 h, and 1, 5, 7 d. In addition, the fine roots of RNAi lines of K^+^ transporter *GhHAK5* (*Gh_D01G1760*) grown in solutions with 2.5 mM K^+^ were sampled at the three-leaf stage. The relative expression of candidate genes was determined by RT-qPCR as above.

### 4.9. Determination of Net K^+^ Uptake Rate in Virus-Induced Gene Silencing (VIGS ) Plants

The Agrobacterium-mediated VIGS was operated according to the method described in Wang et al [41]. Plants of VIGS-candidate genes and VIGS-Ctrl with equal fresh weight were moved into K^+^-starvation solutions (200 mM CaSO_4_, 5 mM MES (pH 5.7) adjusted with Tris) at the three-leaf stage for 48 h. The solutions were refreshed once after 24 h. Each single plant was then transferred to one foil-covered flask containing 50 mL (V_1_) solutions with 0.08 mM K^+^ (C_1_). The volume (V_2_) and K^+^ concentration (C_2_) of solutions were accurately measured after 2, 4, and 6 h (T), then the root fresh weight (FW) of each plant were recorded. The net K^+^ uptake rate (R) was calculated as: R = (V_1_ × C_1_ − V_2_ × C_2_)/(FW × T). Each treatment had four biological replicates.

### 4.10. Statistical Analysis

The data were statistically analyzed using SPSS 21.0 (SPSS Inc., Chicago, IL, USA). Mean values were compared using Duncan multiple comparison procedure at the 1% and 5% level of probability.

## 5. Conclusions

In conclusion, the expression of hundreds of genes changed in response to K^+^ deficiency in cotton roots basically, and more than two thousand genes were associated with feedback regulation of K^+^ uptake and root growth. Figure 9 shows the important genes related to Ca^2+^, ROS, and phytohormone signaling as well as transcription factors (TFs) and transporters, which were involved in these physiological processes. The components of brassinolide (BR) signaling, *GhTCH4* and *GhBKI1s*, were first found to participate in the response to low K^+^ stress and feedback regulation of K^+^ uptake and root growth in plants. In addition, *GhKUP3s*, a member of KUP/HAK/KT family of K^+^ transporters, may play important roles in cotton besides *GhHAK5s*. The two ERF family TFs, GhERF15 and GhESE3, were preliminary demonstrated to negatively regulate K^+^ uptake in cotton through VIGS assay. Finally, the up-regulated genes related to CIPK (CBL-interacting protein kinases) and NO_3_^−^ transporters under low K^+^ possibly contribute to the feedback enhancement of K^+^ uptake in cotton roots.

## Figures and Tables

**Figure 1 ijms-22-03133-f001:**
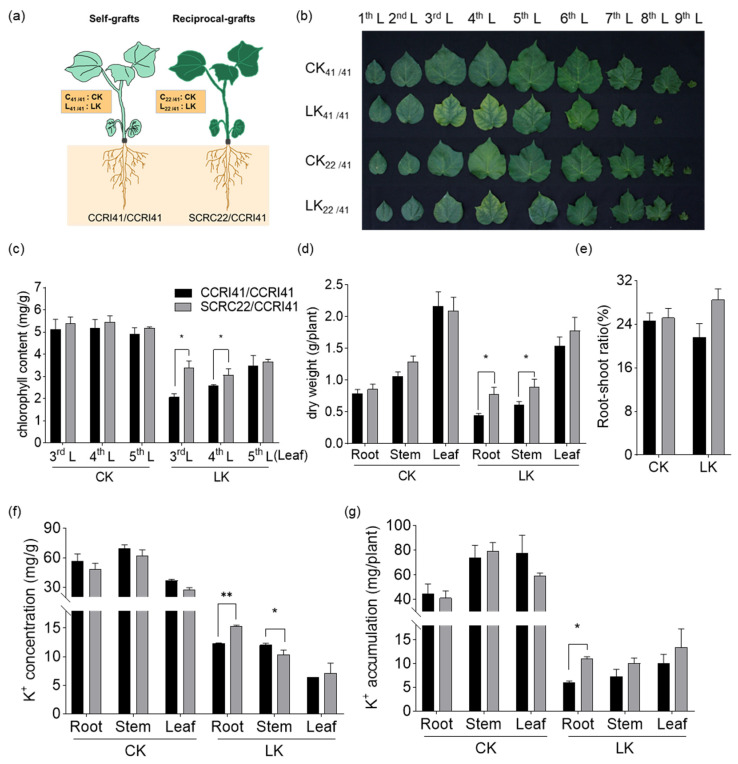
Effects of potassium (K^+^) deficiency on CCRI41 (a K^+^ inefficient cotton variety) self-grafts (CCRI41/CCRI41, scion/rootstock) and its reciprocal grafts (SCRC22/CCRI41) with SCRC22 (a K^+^ efficient variety). Grafting was performed hypocotyl-to-hypocotyl (**a**) when the cotyledons of rootstock just fully expanded. Grafts were subjected to low K^+^ (LK, 0.03 mM K^+^) at the three-leaf stage, with 2.5 mM K^+^ as control (CK). After 16 days, all leaves were photographed (**b**), and the chlorophyll content in the third (3rd L), fourth (4th L) and fifth (5th L) leaves was measured (**c**). The dry weight of roots, stem and leaves was recorded (**d**), and the root-shoot ratio was calculated (**e**). K^+^ concentration (**f**) and K^+^ accumulation (**g**) in roots, stem and leaves were determined or calculated. CK: Control; LK: Low K^+^ treatment. The data are shown as means ± SD of three replicates (*n* = 3); * and ** indicate significant differences at 5% and 1% level, respectively.

**Figure 2 ijms-22-03133-f002:**
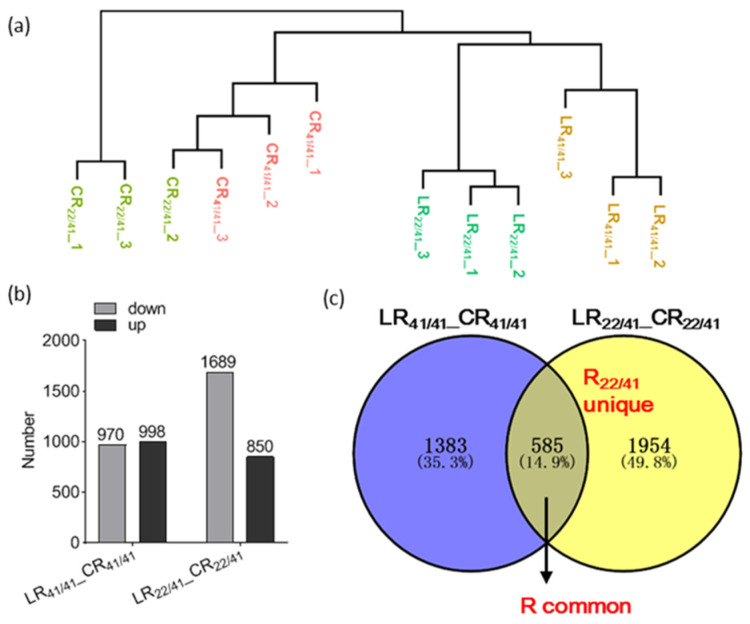
Transcriptome relationships among the roots of CCRI41 self-grafts (CCRI41/CCRI41, scion/rootstock) and reciprocal grafts (SCRC22/CCRI41) under potassium (K^+^) deficiency. (**a**) Cluster dendrogram of the root transcriptomes. (**b**) Number of differentially expressed genes (DEGs) in the roots of CCRI41/CCRI41 and SCRC22/CCRI41. (**c**) Venn diagram showing shared and unique DEGs in both grafts. CR and LR: Roots from CK and LK-treated plants; 41/41: CCRI41/CCRI41; 22/41: SCRC22/CCRI41; 1, 2, and 3: Number of replicates. R common: Common DEGs in the roots of both grafts. R22/41 unique: The DEGs specifically identified in the roots of SCRC22/CCRI41.

**Figure 3 ijms-22-03133-f003:**
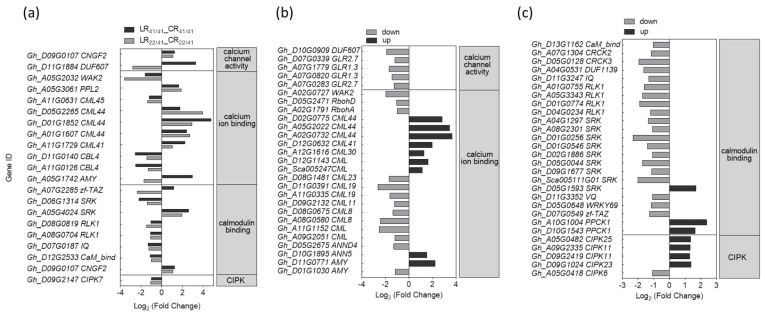
The differentially expressed genes (DEGs) involved in Ca^2+^ (calcium) signaling in R common (**a**) and R_22/41_ unique (**b**,**c**) group. R common: Common DEGs in roots of CCRI41/CCRI41 (scion/rootstock) and SCRC22/CCRI41 under potassium (K^+^) deficiency. R22/41 unique: DEGs specifically identified in roots of SCRC22/CCRI41.

**Figure 4 ijms-22-03133-f004:**
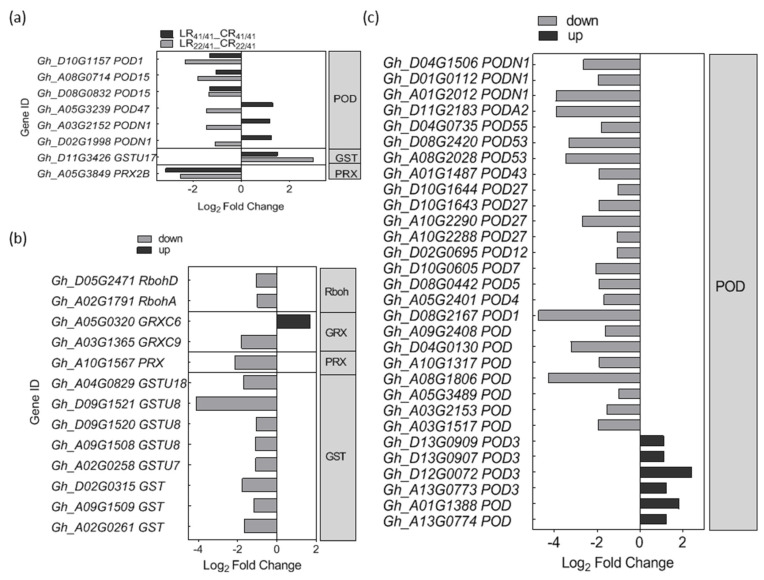
The differently expressed genes (DEGs) involved in ROS (reactive oxygen species) signaling in R common (**a**) and R_22/41_ unique (**b**,**c**) group. R common: Common DEGs in roots of CCRI41/CCRI41 (scion/rootstock) and SCRC22/CCRI41 under potassium (K^+^) deficiency. R22/41 unique: DEGs specifically identified in roots of SCRC22/CCRI41.

**Figure 5 ijms-22-03133-f005:**
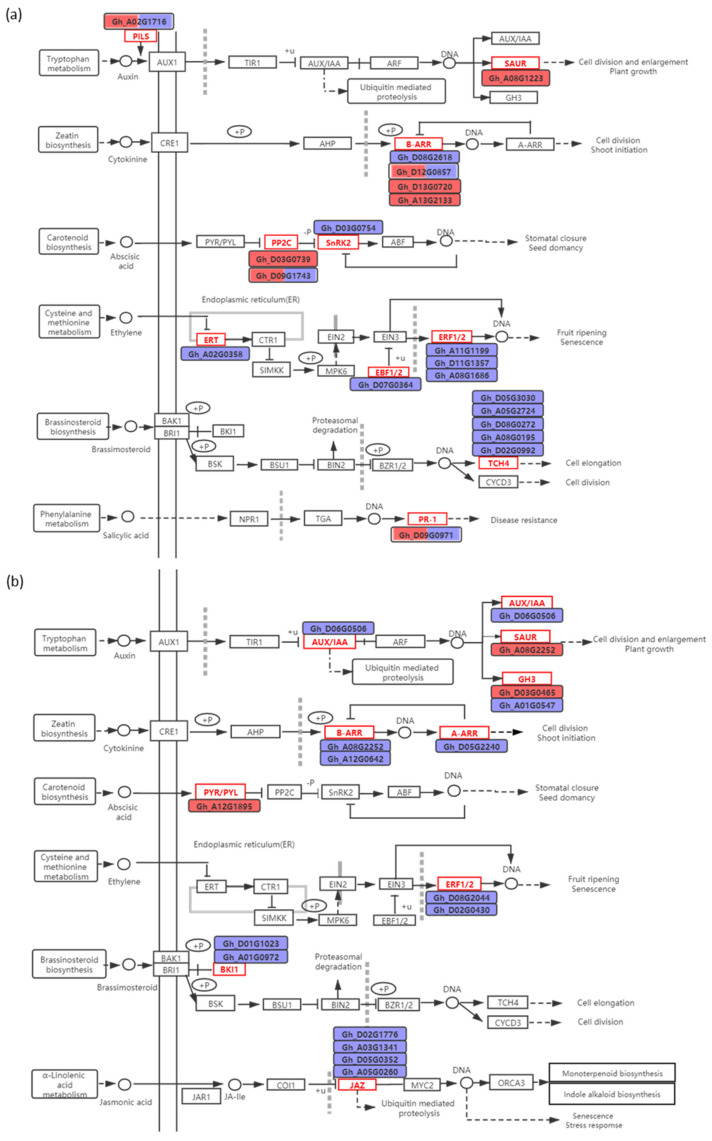
The differentially expressed genes (DEGs) involved in phytohormone signaling in R common (**a**) and R_22/41_ unique (**b**) group. R common: Common DEGs in roots of CCRI41/CCRI41 (scion/rootstock) and SCRC22/CCRI41 under potassium (K^+^) deficiency. R22/41 unique: DEGs specifically identified in roots of SCRC22/CCRI41. Red and purple boxes contain genes that were up- and down-regulated, respectively. The red and blue colored boxes contain genes that were oppositely regulated in both grafts, the left and right boxes were assigned to CCRI41/CCRI41 and SCRC22/CCRI41, respectively. The “DNA” in diagram represents undefined genes.

**Figure 6 ijms-22-03133-f006:**
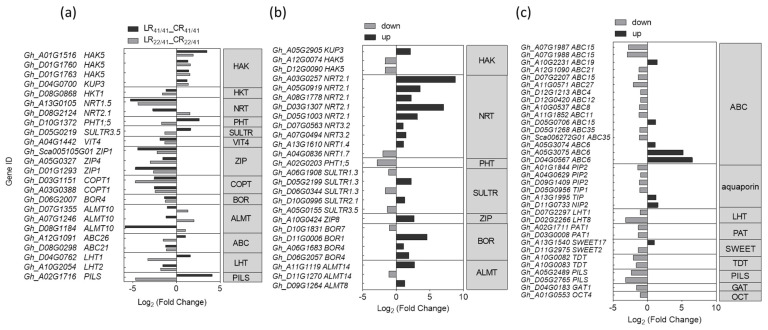
The differentially expressed genes (DEGs) related to transporters in R common (**a**) and R_22/41_ unique (**b**,**c**) group. R common: Common DEGs in roots of CCRI41/CCRI41 (scion/rootstock) and SCRC22/CCRI41 under potassium (K^+^) deficiency. R22/41 unique: The DEGs specifically identified in roots of SCRC22/CCRI41.

**Figure 7 ijms-22-03133-f007:**
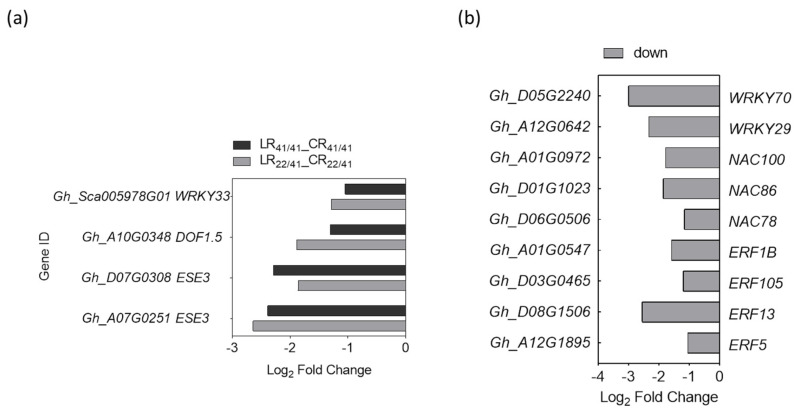
The predicted transcription factors (TFs) that have binding sites in the 2-kb promotor region of the K^+^ transporter gene *GhHAK5* (*Gh_D01G1760*) in R common (**a**) and R_22/41_ unique (**b**) group. R common: Common DEGs in roots of CCRI41/CCRI41 (scion/rootstock) and SCRC22/CCRI41 under potassium (K^+^) deficiency. R22/41 unique: DEGs specifically identified in roots of SCRC22/CCRI41.

**Figure 8 ijms-22-03133-f008:**
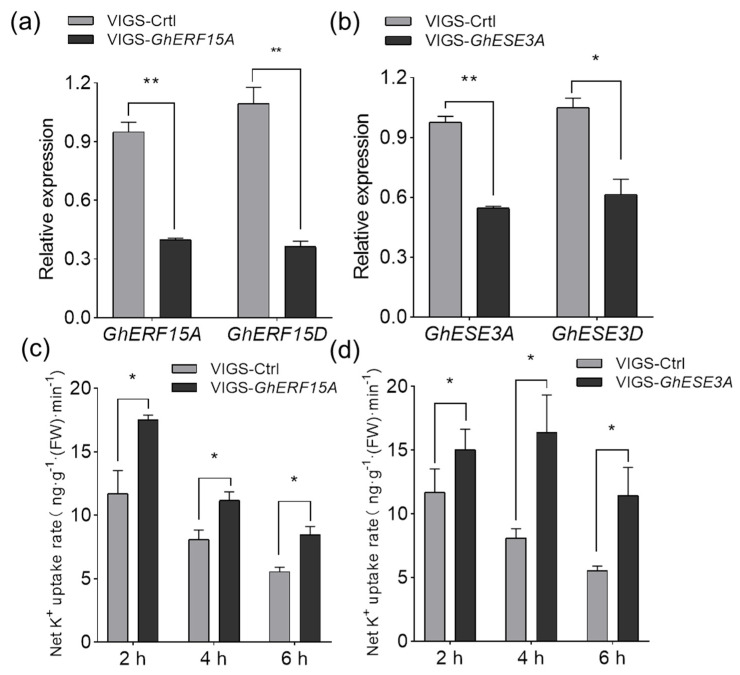
Two members of ethylene response factor (ERF) TFs, *GhERF15* and *GhESE3*, negatively regulate potassium (K^+^) uptake in cotton roots. *GhERF15* and *GhESE3* were silenced in the variety SCRC22 using agrobacterium-mediated virus-induced gene silencing (VIGS) at the cotyledonary stage. The relative expression of *GhERF15* (**a**) and *GhESE3* (**b**) indicates that they were silenced in both A and D subgenome. The seedlings at three-leaf stage were moved into K^+^-starvation solutions for 48 h, then transferred to measuring solution with 0.08 mM K^+^ to determine the net K^+^ uptake rate (**c**,**d**). * and ** indicate significant differences at 5% and 1% level, respectively.

**Figure 9 ijms-22-03133-f009:**
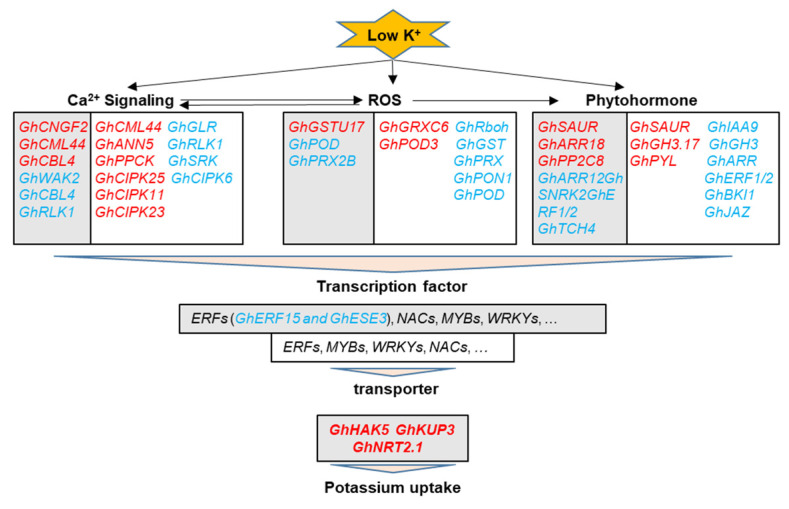
A model of transcription regulation involved in response to potassium (K^+^) deficiency and feedback regulation of K^+^ uptake and root growth in cotton. Genes within the light grey box show the same responses to low K^+^ stress in CCRI41 (a K^+^ inefficient cotton variety) self-grafts (CCRI41/CCRI41, scion/rootstock) and its reciprocal grafts (SCRC22/CCRI41) with SCRC22 (a K^+^ efficient variety); while genes within the white box represent the components involved in feedback regulation of K^+^ uptake and root growth under K^+^ deficiency, they were specifically identified in the roots of SCRC22/CCRI41 or oppositely regulated by low K^+^ in CCRI41/CCRI41 and SCRC22/CCRI41. The red and blue highlighted genes were up- and down-regulated by low K^+^ respectively.

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
