# Peer review of "Transcriptome Analysis Unravels Key Factors Involved in Response to Potassium Deficiency and Feedback Regulation of K+ Uptake in Cotton Roots"

_ijms, 2021, doi:10.3390/ijms22063133_

Round 1

Reviewer 1 Report

Line 26: please, edit: „two ethylene response factor (ERF) family transcription factors“

Line 75: NAPH oxidase itself can not be source of the stress. Please, edit.

Fig 1 E: please, explain how did you do calculation per plants if you use organ? Moreover, it is better to re-calculate per DW, I am not sure contents per organs have biological sense in this case.  Or, at least, show both (per organ and per DW).

Line 155: transcriptome analysis gives you differently expressed genes, but not the mechanism.

Line 218: „Ca2+ sensors play important roles in response to K+ deficiency.“ – to make conckusins about role, you need to change expression of transporters and investigate effcet of this chnages. So far it is only Ca-sensor changes expressions.

Figure 5: what do you mean as DNA in your model? Please, do not forget that hormone effects are dependent from cell status. Auxin can induced both cell expansion (griowth) and cell divison. As an example, root hairs growth and cell division in pericycle (lateral root induction). Please, clarify targets directly in situ.  

Line 413: citation 55 is a mini-review, it is better to cited about effect of DPI, H2O2 etc original papers.

Lines 436-499: please, consider that hormonal signaling act locally. And in the same root auxin signaling can be regulated both up and down in neighbor cells. In this case, in situ data with the cellular resolution are required for conclusions. 

Line 460: negtive- please, change.  

Line 643: please, clarify method of dry weight measure.

Line 720: I would suggest to write „change expression“, but not involved.

Figure 9: please, make a better layout. Colors combinations is not an optimal.

Reviewer 2 Report

In the manuscript ‘Transcriptome analysis unravels key factors involved in response to potassium deficiency and feedback regulation of K+ uptake in cotton roots’ Yang and collaborators performed comparative transcriptome analysis in self-grafts of a K+ inefficient cotton variety and in reciprocal grafts with a K+ efficient variety in order to reveal key factors involved in response to K+ deficiency and feedback regulation of K+ uptake in cotton roots.

Overall, the manuscript is well written, with the exceptions of some confusing phrases that need to be checked (please see below).

I think the abstract needs be improved to make it clear to a wider audience, for example, I suggest to use K+ inefficient variety and K+ efficient variety in first place and put SCRC22/CCRI41 and CCRI41/CCRI41 within parenthesis, instead. The abstract needs a conclusion paragraph so well presented as that of the Conclusion section.

The Introduction presents a good background of the research subject, it introduces the question and the objectives of the study. The experimental design is correct and the methodologies are appropriate to accomplish the objectives, however some points need to be improved (please see below).

The Results section presents clearly the results and the graphics and images are well and clearly presented. The authors make a good discussion of the results and these are compared to the findings of other works. The authors present the conclusions of their work and they also point out further works. The conclusions are justified and supported by the results.

Specific comments

  • Line 13-14: the phrase ‘how its shoot feedback regulates’ is confusing
  • Line 18-20: it’s missing ‘respectively’...
  • Line 169: FPKM - indicate the full name here
  • Line 169-170: reformulate phrase, it is confusing
  • Line 286: remove ‘was also a bit strange’… change to unexpected
  • Line 298: confusing phrase
  • Line 310-311: confusing phrase
  • Line 317-318: confusing phrase
  • Line 324: Sixty seven and 201 … uniformize it
  • Line 382-383: confusing phrase
  • Line 433: assimilates?...
  • Line 501-503: confusing phrase
  • Line 644: indicate how many technical and biological replicates were used
  • Line 649-650: the replicates issue should be clarified… I think the authors used 3 technical replicates each consisting of a pool with 5 biological replicates…
  • Line 651-652: Plant Total RNA Kit (SIGMA, USA) and RNeasy Mini Kit (QIAGEN) are both to extract/isolate/purify RNA, why using 2 different kit to do the same thing?
  • Line 653-654: RNA integrity and RNA quality is the same thing
  • Line 655: remove qualified, it can be replaced by high-quality RNA
  • Line 685: qRT-PCR change to RT-qPCR
  • Line 686: explain here the aim of this analysis?
  • Line 688: qRT-PCR change to RT-qPCR
  • Line 690: to calibrate gene expression change to normalize gene expression levels
  • Line 690: the authors should include PCR run thermocycling conditions
  • Line 691: qRT-PCR change to RT-qPCR
  • Line 692: the authors should include in table S3 the amplicon length and annealing temperature
  • Line 697: reformulate the phrase
  • Line 698: the title of the 4.8. subsection and the methodology description do not correspond to each other
  • Line 714: specify whether technical or biological replicates
  •  

Reviewer 3 Report

The study by Yang et al identifies the changes in transcript abundance in cotton leaves, roots and shoots in response to K deficiency. This is a nice study with a clear biological question in mind. I liked the way the authors have conducted the experiments and provided a good overview of their methods, results and discussion.

Few minor comments are given below,

  • Some of the text for figure 1 and 5 is hard to read. Can authors please look into it?
  • Can authors present the root-shoot ratios as a graph? Authors need to indicate what is C and L (Figure 1b) in the figure legend as well. In fact, it would be good to replace “C” with “CK” and “L” with “LK“similar to figure S1.
  • Can authors also show the gene differences between CR41/4 and CR22/41 that could suggest the extent of genetic variation between the two genotypes?
  • There are some inconsistencies between Figure 3 legend and the text. For example. The figure legend describes Figure 3 (a) as R common group but the text suggests “R common group” as figure 3b.
  • Mostly, strigolactones (SL) are produced because of K deficiency, it is interesting to see that cotton roots did not show significant changes in the gene transcripts associated with SL production/perception in response to K deficiency. A recent study have shown the role of SL in cotton fibre development (https://doi.org/10.3389/fpls.2019.00087) and previously SL were found in cotton roots (Cook et al., m1966). It will be good to get the authors point of view on SL functions during K deficiency in cotton roots in the discussion section.

Author Response

Response to Reviewer 3 Comments

Point 1: Some of the text for figure 1 and 5 is hard to read. Can authors please look into it?

Response: We thank this suggestion, and have done accordingly.

The text of Figure 1 was revised as following:

Grafting was performed hypocotyl-to-hypocotyl (Figure 1a) when the cotyledons of rootstock just fully expanded. The low-K+ stress (LK, 0.03 mM K+) was applied at three-leaf stage. After 16 days, we observed that the interveinal chlorosis on the third and fourth leaves was more severe in self-grafts of CCRI41/CCRI41 than in reciprocal-grafts of SCRC22/CCRI41 (Figure 1b). Consistently, CCRI41/CCRI41 showed lower chlorophyll content in these two leaves under low K+ condition (Figure 1c). Moreover, the biomass of roots, stem and leaves reduced to varying extents under K+ deficiency, and CCRI41/CCRI41 was impacted more strongly than SCRC22/CCRI41, especially in terms of root biomass (Figure 1d). The root-shoot ratio of CCRI41/CCRI41 decreased by 3.8% under low K+ condition, which was similar to previous reports [41]. However, the root-shoot ratio of SCRC22/CCRI41 increased by 3.7% (Figure 1e). Furthermore, the SCRC22/CCRI41 showed higher K+ concentration and accumulation in roots and leaves than CCRI41/CCRI41 under low K+ stress, and stored more K+ in stem than CCRI41/CCRI41 mainly due to higher dry weight (Figure 1f, g). These results suggest that SCRC22 scion could feedback enhance root growth and K+ uptake on CCRI41 rootstock.

The text of Figure 5 was revised as following:

The 19 and 16 DEGs related to phytohormone were found in R common and R22/41 unique group, respectively (Figure 5). The similarly regulated (both up- or both down-) DEGs in both grafts were associated with auxin (1 up-regulated GhSAUR gene), CTK (1 down-regulated GhARR12 gene and 2 up-regulated GhARR18 genes), ABA (1 down-regulated GhSNRK2 gene and 1 up-regulated GhPP2C8 gene), ethylene (5 down-regulated genes, GhETR2, GhEBF1, two GhERF1B genes, and GhERF15A), and BR (5 down-regulated GhTCH4 genes) (Figure 5a).

The one DEG each associated with auxin, CTK, ABA and SA were oppositely regulated (one up- and the other down-) in CCRI41/CCRI41 and SCRC22/CCRI41 (Figure 5a). For the R22/41 unique group, the DEGs related to auxin (GhIAA9 and GhGH3), CTK (GhARR2, GhARR12 and GhARR9), ethylene (GhERF15D and GhERF1B), BR (two GhBKI1 genes), and JA (four GhJAZ genes) were significantly down-regulated, whereas the two auxin-responsive genes (GhGH3.17 and GhSAUR) and one ABA receptor gene (GhPYL) were significantly up-regulated (Figure 5b).

Point 2: Can authors present the root-shoot ratios as a graph? Authors need to indicate what is C and L (Figure 1b) in the figure legend as well. In fact, it would be good to replace “C” with “CK” and “L” with “LK“similar to figure S1.

Response: We have added the graph of root-shoot ratios in figure 1. We agree with this suggestion that replace “C” with “CK” and “L” with “LK” in Figure 1b, and have added “CK: control treatment; LK: low K+ treatment.” as the figure legend in line 164.

Point 3: Can authors also show the gene differences between CR41/4 and CR22/41 that could suggest the extent of genetic variation between the two genotypes?

Response: Thanks for this suggestion. However, we used the roots of self-grafts (CCRI41/CCRI41, scion/rootstock) and its reciprocal grafts (SCRC22/CCRI41) as materials, so there were no genetic differences between their rootstock.

Point 4: There are some inconsistencies between Figure 3 legend and the text. For example. The figure legend describes Figure 3 (a) as R common group but the text suggests “R common group” as figure 3b.

Response: We thank this question. After checking the figure legends and the text of figure 2 and figure 3, we found that there was a mistake in the text of figure 2, and have revised it in line 199.

Point 5: Mostly, strigolactones (SL) are produced because of K deficiency, it is interesting to see that cotton roots did not show significant changes in the gene transcripts associated with SL production/perception in response to K deficiency. A recent study have shown the role of SL in cotton fibre development (https://doi.org/10.3389/fpls.2019.00087) and previously SL were found in cotton roots (Cook et al., m1966). It will be good to get the authors point of view on SL functions during K deficiency in cotton roots in the discussion section.

Response: We welcome this suggestion. Although strigolactones (SLs) was first isolated from cotton root exudates and is involved in the regulation of cotton fiber development, we did not identify DEGs related to SLs in this study. So, we think it is not very proper to discuss it in the Discussion section, and hope to be understood by reviewer.

Round 2

Reviewer 1 Report

The authors provided all answers to the comments.
 Only small points still need to be done:
The gene DEG and mechanism: authors used whole organ containing different cell types, and under K-deficiency, these cell types respond differently. The cortex cell of the root may respond oppositely as pericycle, and you will not see the change in DEG. A direct link between DEG and mechanism exists only in the homogenous cell population when all cells respond in the same direction.
Figure 5: auxin also the main regulator of cell division, not cell expansion.
 And plant growth is not a cell expansion. it is cell division and thereafter expansion.
Please, make corrections.
Moreover, do you mean seeds, not srrds for ABA?

It will be also great to add at least few words in discussion about epigenetic because K-dieficeincy have a significant effect on plant epigenetic status. But, again, with cell type/position specific manner. 
